# Deep phosphoproteomics of *Klebsiella pneumoniae* reveals HipA-mediated tolerance to ciprofloxacin

Payal Nashier[1], Isabell Samp[2], Marvin Adler[2], Fiona Ebner[2], Lisa Thai Lê[2], Marc Göppel[2], Carsten Jers[3], Ivan Mijakovic[3,4], Sandra Schwarz[2]*, Boris Macek[1]*

1 Proteome Center Tübingen, Institute of Cell Biology, University of Tübingen, Tübingen, Germany,
2 Interfaculty Institute of Microbiology and Infection Medicine Tübingen, Institute of Medical Microbiology and Hygiene, University of Tübingen, Tübingen, Germany, 3 The Novo Nordisk Foundation, Center for Biosustainability, Technical University of Denmark, Kongens Lyngby, Denmark, 4 Systems and Synthetic Biology Division, Department of Life Sciences, Chalmers University of Technology, Gothenburg, Sweden

* sandra.schwarz@med.uni-tuebingen.de (SS); boris.macek@uni-tuebingen.de (BM)

## Abstract

*Klebsiella pneumoniae* belongs to the group of bacterial pathogens causing the majority of antibiotic-resistant nosocomial infections worldwide; however, the molecular mechanisms underlying post-translational regulation of its physiology are poorly understood. Here we perform a comprehensive analysis of *Klebsiella* phosphoproteome, focusing on HipA, a Ser/Thr kinase involved in antibiotic tolerance in *Escherichia coli*. We show that overproduced *K. pneumoniae* HipA (HipA$_{kp}$) is toxic to both *E. coli* and *K. pneumoniae* and its toxicity can be rescued by overproduction of the antitoxin HipB$_{kp}$. Importantly, HipA$_{kp}$ overproduction leads to increased tolerance against ciprofloxacin, a commonly used antibiotic in the treatment of *K. pneumoniae* infections. Proteome and phosphoproteome analyses in the absence and presence of ciprofloxacin confirm that HipA$_{kp}$ has Ser/Thr kinase activity, autophosphorylates at S150, and shares multiple substrates with HipA$_{ec}$, thereby providing a valuable resource to clarify the molecular basis of tolerance and the role of Ser/Thr phosphorylation in this human pathogen.

## Author summary

*Klebsiella pneumoniae* is a bacterial pathogen that causes hospital-acquired infections in immunocompromised patients, often becoming resistant or tolerant to multiple antibiotics. These bacteria are becoming increasingly difficult to treat due to the relapse of infection by multidrug tolerant persister cells. Our research focuses on characterizing the kinase HipA in *K. pneumoniae*, which is known to be involved in antibiotic persistence in *E. coli*. We studied HipA-dependent protein phosphorylation in *K. pneumoniae* to understand the mechanism of persistence. We found that HipA induces antibiotic tolerance to ciprofloxacin treatment but not gentamicin. To the best of our knowledge, this is the first study that addresses post-translational regulation in *K. pneumoniae* and connects protein phosphorylation with drug tolerance in this important human pathogen. This study will

**Data Availability Statement:** The mass spectrometry proteomics data have been deposited to the ProteomeXchange Consortium via the PRIDE [63] partner repository with the dataset

identifier PXD051521. All data needed to evaluate the conclusions in this paper are available in the main text or the supplemental information.

**Funding:** This work has received funding from the European Union's Horizon 2020 research and innovation programme under Marie Sklodowska-Curie grant agreement No. 955626 (P.N., B.M. and I.M.), German Research Foundation (DFG) CMFI Cluster of Excellence (EXC-2124); grant number EXC-2124/1-06.006 (S.S. and B.M.) and TRR 261-Project-ID 398967434 (B.M.), and Novo Nordisk Foundation grant NNF20CC0035580 (I.M.). P.N. received a salary support from grant no. 955626. The funders had no role in study design, data collection and analysis, decision to publish, or preparation of the manuscript.

**Competing interests:** The authors have declared that no competing interests exist.

be a valuable resource for both microbiologists and systems biologists in better understanding of persister infections.

# Introduction

*Klebsiella pneumoniae* is a Gram-negative, extended-spectrum β-lactamase (ESBL)-producing, pathogenic bacterium that causes hospital-acquired infections in immunocompromised patients but also community-acquired infections in healthy individuals [1]. *K. pneumoniae* poses a severe risk, causing potentially deadly infections like bloodstream infections and pneumonia, especially in healthcare environments with vulnerable patients and medical devices. The bacteria belong to the "ESKAPE" group of antimicrobial-resistant and virulent pathogens (*Enterococcus faecium*, *Staphylococcus aureus*, *Klebsiella pneumoniae*, *Acinetobacter baumannii*, *Pseudomonas aeruginosa*, *Enterobacter spp.*), causing the majority of nosocomial infections worldwide [2]. Furthermore, the WHO considers *K. pneumoniae* as a Priority I pathogen for the development of novel antibiotics due to the escalating resistance against antibiotics including last-resort antimicrobials [3,4].

Antibiotic tolerance, defined as the ability of a whole bacterial population to survive a transient antibiotic exposure to concentrations much higher than the minimum inhibitory concentration (MIC) is an alternative mechanism enabling evasion of antibiotic therapy and causing relapse of infections. Extended exposure to an antibiotic, as opposed to a higher dosage of the drug, can cause the same amount of killing in tolerant and sensitive cells [5]. Antibiotic tolerance plays a significant role in shaping the evolutionary dynamics of bacterial populations subjected to repeated antibiotic treatments. Notably, it was reported that antibiotic tolerance promotes the subsequent emergence of antibiotic resistance [6,7]. The molecular mechanisms of tolerance are also linked with time-dependent antibiotic persistence, which emerges in a heterogenous population of clonal bacteria when only a subpopulation develops tolerance. These time-dependent persisters exhibit a biphasic killing curve characterized by slow growth and are insensitive to substantially high concentrations of antibiotics [5]. Therefore, persister cells are phenotypic variants of the normal sensitive population of cells with the ability to survive high concentrations of antibiotic exposure [8,9].

*K. pneumoniae* has the ability to produce persister cells, and their development was shown to be strongly stimulated by stationary-phase related environmental cues and sublethal concentrations of antibiotics [10]. Exposure to various bactericidal antibiotics commonly employed in the treatment of *K. pneumoniae* infections has revealed the presence of multi-drug-tolerant persister cells in both laboratory and clinical strains, as determined through time-dependent killing curves [11]. Additionally, persister cells have been identified in clinical isolates from individuals experiencing recurring bloodstream infections, demonstrating genomic alterations in relapsed isolates that evolved within the host [12]. However, the mechanisms underlying the formation of these multidrug-tolerant persister cells in *K. pneumoniae* are understudied, although this knowledge could be important for managing chronic infections more efficiently and devising strategies to eliminate persisters.

HipA is a well-characterized protein in *Escherichia coli* (HipA$_{ec}$) that was previously shown to induce persistence and be involved in antibiotic tolerance [13,14]. Acting as its antitoxin, HipB, a DNA-binding transcriptional regulator, binds to HipA, forming a HipBA protein complex that represses its own operon under normal conditions [15]. Degradation of HipB by Lon proteases, upregulated during stress conditions, results in the release and activation of HipA [16]. HipA$_{ec}$ is a Ser/Thr kinase that phosphorylates glutamyl-tRNA synthetase (GltX)

causing accumulation of uncharged Glu-tRNA. This in turn halts translation, leading to the activation of the stringent response and induction of persistence by RelA-mediated synthesis of the alarmone ppGpp [17,18]. Phosphoproteomic study based on over-expression of $hipA_{ec}$ has shown that $HipA_{ec}$ phosphorylates multiple proteins in addition to the well-described substrate GltX [19]. A more recent bioinformatics study showed that HipA-like kinases are abundant across different bacterial species, and revealed the presence of a homolog in *K. pneumoniae* [20]. Due to the association of *hipA*-related genes to antibiotic tolerance and persistence, we hypothesized that the HipA-homolog in *K. pneumoniae* may have a similar function to $HipA_{ec}$ and set out to investigate its activity and targets in this important human pathogen.

For the molecular characterization of the HipA-homolog in *K. pneumoniae*, we designed a series of experiments to analyze the effect of $hipA_{kp}$ overexpression in *E. coli* and *K. pneumoniae* cells. Using quantitative mass spectrometry-based phosphoproteomics [21], we measured and analyzed the phosphoproteome of $HipA_{kp}$-overproducing cells to identify the potential substrates of $HipA_{kp}$ and also assessed its effect on antibiotic tolerance. Here we show that $hipA_{kp}$ overexpression is toxic to the cells and toxicity can be partially rescued by $hipB_{kp}$ overexpression. We confirmed that $HipA_{kp}$ is a Ser/Thr kinase that autophosphorylates at S150 and T158 and also phosphorylates GltX at S239 in both *E. coli* and *K. pneumoniae*. In addition to GltX, we discovered numerous additional putative substrates of the kinase, involved in translation, transcription, cell division, and central metabolism. Finally, we found that overexpression of $hipA_{kp}$ leads to tolerance against the fluoroquinolone antibiotic ciprofloxacin, thereby connecting the function of this kinase with antibiotic tolerance in *Klebsiella*.

## Results

### 1.1 The *hipB/A* operon is conserved across *Klebsiella* and other Gram-negative bacteria

The *hipB/A* operon is established as one of the main drivers of antibiotic tolerance in *E. coli*; therefore, we first compared the amino acid sequence of HipB/A with its putative homolog in *K. pneumoniae*. Pairwise sequence alignment using BLASTp showed that $HipA_{kp}$ and $HipB_{kp}$ shared 69% and 56% of sequence identity with $HipA_{ec}$ and $HipB_{ec}$, respectively (**Figs 1A, S1A and S1B**). The alignment of the $HipA_{kp}$ structure as predicted by AlphaFold [22,23] with the experimentally determined $HipA_{ec}$ structure [24] showed a high degree of conservation (RMSD 0.444 Å), including within the ATP and $Mg^{2+}$ ion binding pockets important for kinase activity (**Fig 1B**). Furthermore, several residues known to be essential for kinase activity were conserved between $HipA_{kp}$ and $HipA_{ec}$, such as the autophosphorylation site S150, the catalytic residue D309, the residue L181 involved in ATP-binding, and the residues N314 and D332 involved in $Mg^{2+}$ binding [13] (**S1A Fig**). Alignment of the AlphaFold-predicted structure of $HipB_{kp}$ with the experimentally determined structure of $HipB_{kp}$ [24] showed a similar degree of conservation (RMSD 0.405 Å) (**S1C Fig**).

Protein BLAST analysis showed that the $HipA_{kp}$ is conserved and present in different species of *Klebsiella*, with the mean percent identity varying from 95% to 99% in different isolates of the same genus, species and subspecies (**Fig 1C**). For further visualization of the presence of $HipA_{kp}$ across all organisms, we plotted the percentage of sequence identity of $HipA_{kp}$ homologs from the top 5,000 protein BLAST hits and determined that the sequence identity reaches up to 70% in many Gram-negative bacteria such as *E. coli* and bacteria belonging to the genus *Shigella*, *Serratia* and *Salmonella*. Some of the genera were under-represented due to the large number of hits originating from *Klebsiella* and *E. coli* (**S1D Fig**). Combined, these results

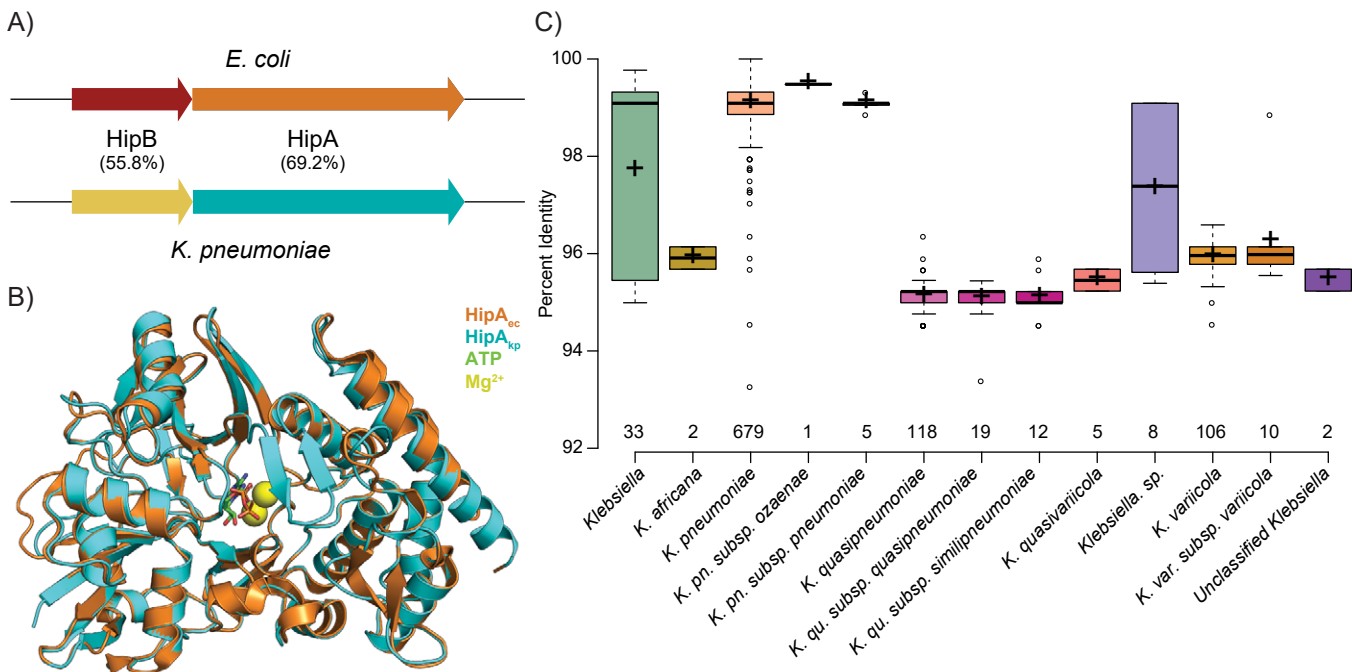

**Fig 1. Bioinformatic analysis of *hipA*$_{kp}$ and *hipB*$_{kp}$. A)** Schematic representation of the *hipB/A* operon in *K. pneumoniae* and *E. coli*. The numbers indicate the percent identity of HipA and HipB between the species. **B)** Alignment of AlphaFold predicted structure of HipA$_{kp}$, UniProt ID: A6T928 (cyan), with the experimentally determined structure of HipA$_{ec}$, PDB ID: 3DNT (orange), together with ATP (sticks), and Mg$^{2+}$ molecules (yellow spheres), in the conserved pocket. **C)** Distribution of percentage of sequence identity of HipA$_{kp}$ homologs within the *Klebsiella* genus and the number of hits obtained upon analyzing the top 1,000 results from protein BLAST of HipA$_{kp}$ protein with all the default settings except limiting the organism search to "Klebsiella (taxid:570)" as "Organism" in Standard settings.

showed high structural similarity of HipB/A between *K. pneumoniae* and *E. coli*, as well as conservation of HipA among many Gram-negative bacteria.

## 1.2 Overproduction of HipA$_{kp}$ is toxic to the cell and can be counteracted by HipB$_{kp}$

To investigate the role of *hipA*$_{kp}$, we first analyzed its effect on the growth and viability of *E. coli* cells in LB medium. As previously reported for *hipA*$_{ec}$, overexpression of *hipA*$_{kp}$ was expected to be toxic to the cells. Therefore, we ectopically expressed *hipA*$_{kp}$ in *E. coli* under the control of an arabinose-inducible promoter with an optimized Shine-Dalgarno sequence [25]. We observed that overexpression of *hipA*$_{kp}$ was highly toxic to *E. coli* cells, resulting in reduction of their growth after 1 h post-induction by three-fold and survival by 2.5-fold, as measured by optical density (OD$_{600nm}$) and colony forming units (CFU), respectively (**Fig 2A**). To determine whether the overproduction of the putative antitoxin HipB$_{kp}$ counteracted the activity of *hipA*$_{kp}$ in *E. coli*, we simultaneously overexpressed *hipA*$_{kp}$ and *hipB*$_{kp}$ from different plasmids. Compared to the growth of *E. coli* overexpressing only *hipA*$_{kp}$, simultaneous overproduction of *hipB*$_{kp}$ restored the growth of *hipA*$_{kp}$-overexpressing *E. coli* and therefore reversed the *hipA*$_{kp}$-related toxic phenotype (**Fig 2B**). We therefore concluded that overproduced HipA$_{kp}$ and HipB$_{kp}$ act as a canonical toxin/antitoxin pair in *E. coli*.

We next overexpressed *hipA*$_{kp}$ in the native *K. pneumoniae* background. To this end, we used the *K. pneumoniae* isolate ATCC13883 (wild-type, WT), which harbors the *hipB/A* operon on the chromosome. First, we generated an unmarked *hipA*$_{kp}$ gene (Δ*hipA*) deletion mutant and assessed the impact of the deletion on the growth and viability of *K. pneumoniae*

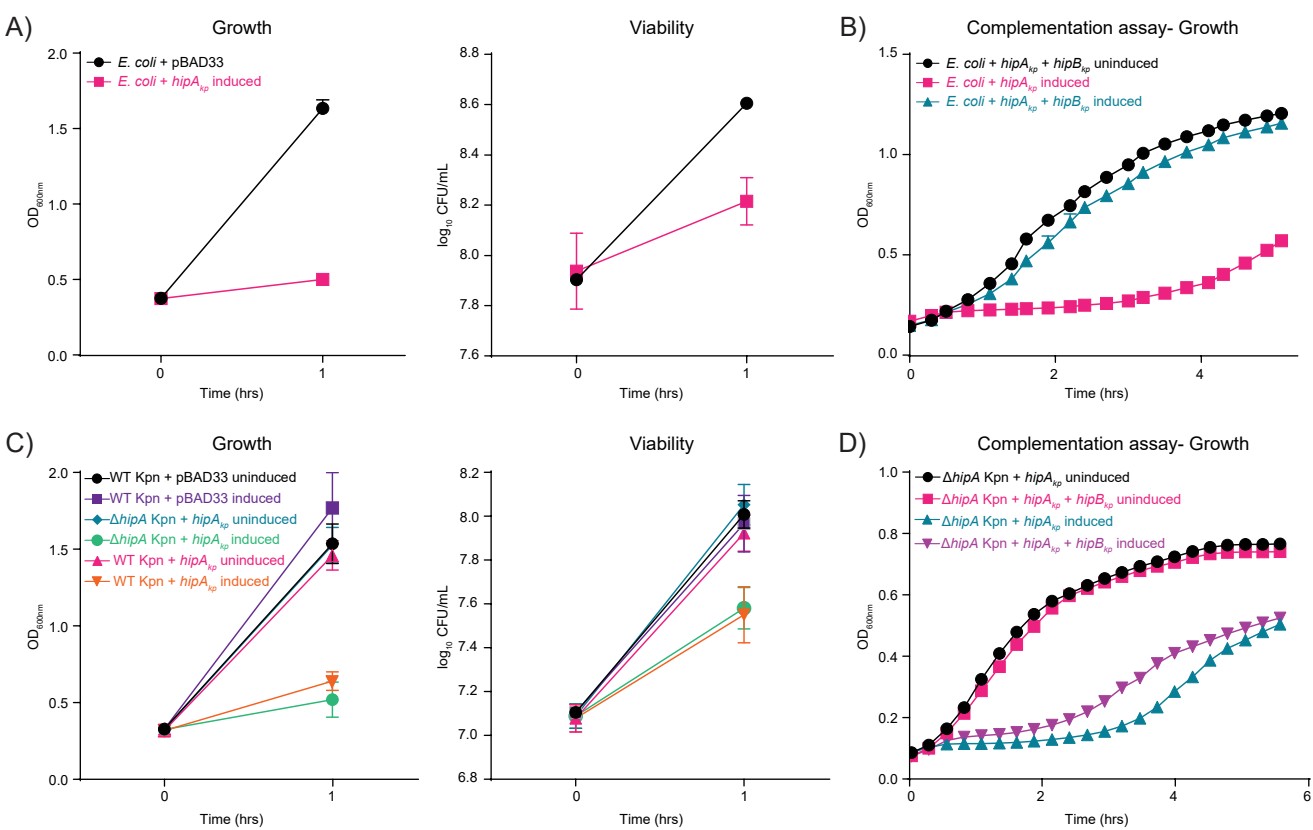

**Fig 2. Effect of overexpression of *hipA_kp* on WT *E. coli* and WT & Δ*hipA K. pneumoniae* cells.** A) Growth and viability of *E. coli* containing the empty pBAD33 vector and pBAD33::*hipA_kp*. The expression of *hipA_kp* is under the control of the arabinose-inducible promoter. After the cells reached an $OD_{600nm}$ of 0.3, expression of *hipA_kp* was induced with 0.2% arabinose for 1 h. Growth was monitored by absorbance measurements at $OD_{600nm}$ and viability was determined by CFU quantification. Bacteria were harvested at 1 h post-induction for proteome and phosphoproteome analysis. B) Growth of *E. coli* carrying pBAD33::*hipA_kp* alone or together with plasmid pGOOD::*hipB_kp*, in which *hipB_kp* is under the control of an IPTG-inducible promoter. Overnight cultures of the bacteria were used to inoculate the cultures for the assay at 0.08 $OD_{600nm}$ in medium containing 0.2% arabinose and 1 mM IPTG. As a control, one set of samples was left uninduced and one strain carrying and expressing only *hipA_kp*. The growth was followed via $OD_{600nm}$ measurements for 5 h in a plate reader (Tecan). C) Growth and viability of WT *K. pneumoniae* harboring the empty pBAD33 plasmid and pBAD33::*hipA_kp* and Δ*hipA* containing pBAD33::*hipA_kp* with expression under the control of arabinose-inducible promoter. The strains were grown in LB Lennox and induced with 0.2% arabinose at $OD_{600nm}$ 0.3 for 1 h and growth was followed by optical density at $OD_{600nm}$ and CFU for viability on plate. Samples were harvested at 1 h post-induction for proteome and phosphoproteome analysis. D) Growth of Δ*hipA K. pneumoniae* carrying pBAD33::*hipA_kp* and pGOOD::*hipB_kp* in which expression of *hipA_kp* and *hipB_kp* is under the control of arabinose-inducible and IPTG-inducible promoters, respectively. Cultures were started at 0.05 $OD_{600nm}$ and grown till $OD_{600nm}$ of 0.3 and induced with 0.2% arabinose and 1 mM IPTG. Uninduced conditions served as controls. The growth was followed via optical density for approximately 5 h in a plate reader. All the plots are representative of mean values ± SD of three independent experiments.

by measuring the optical density ($OD_{600nm}$) and CFU levels. We observed that the growth of the Δ*hipA* mutant was almost identical to that of the WT in LB over a 24 h incubation period. Likewise, the deletion of *hipA* did not affect the survival of *K. pneumoniae* in the exponential phase and resulted in a slight reduction of CFU/ml in the stationary phase (**S2A Fig**). We next overexpressed *hipA_kp* from the pBAD33 vector in both WT and Δ*hipA K. pneumoniae*. The overproduction of HipA_kp decreased growth and viability by three-fold at 1 h post-induction (**Fig 2C**). We tested if the overexpression of *hipB_kp* in *K. pneumoniae* WT + *hipA_kp* and Δ*hipA* + *hipA_kp* could counter the toxic effect of *hipA_kp* expression. We observed that bacteria expressing both *hipA_kp* and *hipB_kp* grew better than Δ*hipA* + *hipA_kp* indicating that HipB_kp partially rescued the toxic effect of HipA_kp (**Fig 2D**). These results point to a similar role of the antitoxin HipB in *K. pneumoniae* and *E. coli* cells.

### 1.3 HipA$_{kp}$ has Ser/Thr kinase activity and phosphorylates multiple substrates

Based on the sequence similarity of HipA$_{kp}$ to HipA$_{ec}$ and the presence of conserved kinase and ATP-binding domains (**S1A Fig**)**,** we postulated that HipA$_{kp}$ has kinase activity. In order to identify its putative substrates, we overexpressed *hipA$_{kp}$* in *E. coli* and performed quantitative phosphoproteome analysis using liquid chromatography coupled to tandem mass spectrometry (LC-MS/MS). At the proteome level, we performed a one-sample t-test with FDR<0.05 to prepare a volcano plot, based on the log$_2$ ratio of WT *E. coli* with HipA$_{kp}$ induced and empty vector (**S2B Fig**), which showed that overproduced HipA$_{kp}$ did not affect the regulation of other proteins. We measured a more than eight-fold increase in HipA$_{kp}$ in the cells upon *hipA$_{kp}$* overexpression for 1 h in comparison with the empty pBAD33 vector control in both replicates (**S2C Fig and Tab A in S1 Dataset**). These results confirmed the efficiency of the *hipA$_{kp}$* overexpression strategy.

At the phosphoproteome level, we identified 317 phosphorylation sites on 189 proteins in *E. coli* + *hipA$_{kp}$* 1 h post-induction, with an excellent correlation between the replicates (Pearson's correlation coefficient value of 0.91) (**S2D Fig and Tab B in S1 Dataset**). Upon HipA$_{kp}$ overproduction, we reproducibly detected increased phosphorylation of multiple substrates; among them was GltX, which showed a more than 16-fold increase in phosphorylation at position S239 in all replicates. The direct comparison of phosphoproteome ratios normalized to the proteome and unnormalized confirmed that most of the phosphorylation sites that were more than four-fold up-regulated, remain unaffected by normalization except for HipA$_{kp}$ autophosphorylation sites (**S2E Fig and** Tab A in **S1 Dataset**). The data indicate that HipA$_{kp}$ has kinase activity and GltX is its major substrate in *E. coli*.

To investigate HipA$_{kp}$ activity and identify its potential targets in the native organism *K. pneumoniae*, we induced expression of *hipA$_{kp}$* from pBAD33 in WT and Δ*hipA K. pneumoniae* in LB medium for 1 h and performed LC-MS/MS analysis. The reproducibility between the proteome and phosphoproteome data of three replicates in both WT and Δ*hipA* backgrounds with HipA$_{kp}$ overproduction was high (overall Pearson's correlation coefficient >0.7) (**S3A and S3B Fig**). One sample t-test was performed based on the log$_2$ proteome ratios from three replicates. The results revealed a more than 16-fold increase in HipA$_{kp}$ levels in the WT background (**Fig 3A**) and Δ*hipA* (**Figs 3B and S3C**) as compared to bacteria harboring the empty vector (**Tab C in S1 Dataset**). At the phosphoproteome level, we identified a total of 747 phosphorylation sites, belonging to 417 phosphoproteins (**Tab D in S1 Dataset**). These phosphorylation sites were filtered for localization probability of >0.75. Upon HipA$_{kp}$ overproduction, we reproducibly detected increased autophosphorylation of HipA on S150 and T158 by more than 16-fold, as well as phosphorylation of GltX on S239, also by 16-fold, in both WT (**Fig 3C**) and Δ*hipA K. pneumoniae* (**Fig 3D**). The comparison of Δ*hipA* and WT *K. pneumoniae* overexpressing *hipA$_{kp}$* phosphoproteome showed a high correlation of 0.83, with several phosphorylation sites more than four-fold upregulated compared to empty vector control (**S3D Fig**)**.** Up-regulation of these additional sites indicates that HipA$_{kp}$ acts on multiple substrates that may play a role in the toxic phenotype. All the phosphorylation sites were normalized to the proteome which showed that most of the up-regulated phosphorylation sites remain unaffected by normalization except HipA autophosphorylation sites in WT and Δ*hipA K. pneumoniae* (**Figs 3E, 3F, S3E and S3F**).

### 1.4 HipA$_{kp}$ overproduction in *K. pneumoniae* leads to increased tolerance to ciprofloxacin, but not to gentamicin

We next investigated the effect of HipA$_{kp}$ overproduction on the growth and survival of *K. pneumoniae* in the presence of gentamicin or ciprofloxacin, antibiotics commonly used in

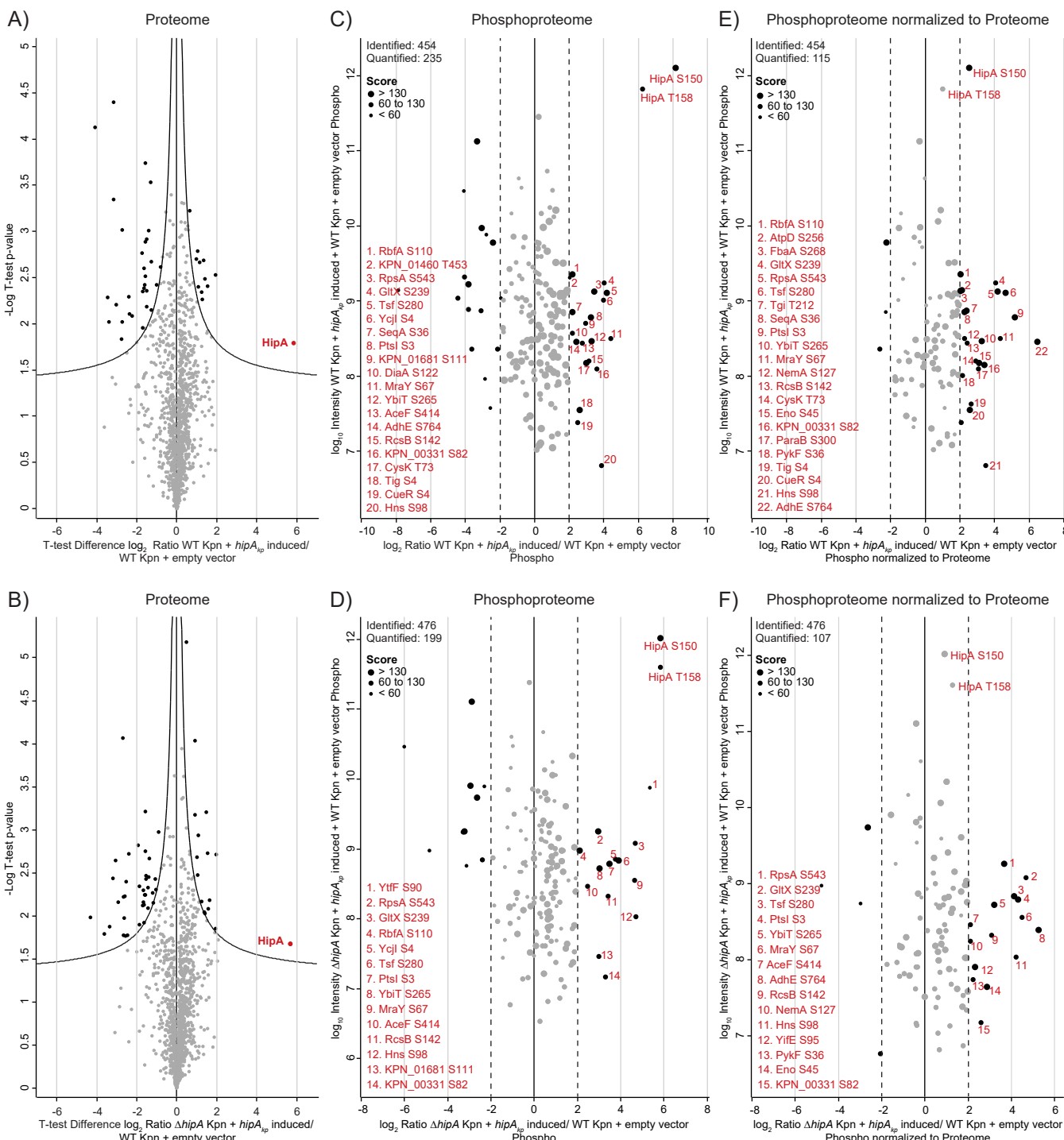

**Fig 3. Effect of overexpression of *hipA*$_{kp}$ on the proteome and phosphoproteome of WT and Δ*hipA* K. pneumoniae. A and B)** Volcano plots showing differential expression to compare the proteome of *K. pneumoniae* cells overproducing HipA$_{kp}$ with the empty vector control. Student's one sample t-test was performed for n = 3 biological replicates. The x-axis represents log$_2$ fold changes based on the indicated ratio and the y-axis represents negative log$_{10}$ of the Benjamini-Hochberg corrected (FDR<0.05) t-test p-value. The curve lines indicate confidence levels with p-value = 0.05. Proteins with a p-value of < 0.05 are indicated by black dots and represent the significantly up and down-regulated proteins. **C and D)** Scatter plot showing the distribution of quantified phosphorylation sites upon *hipA*$_{kp}$ overexpression in WT and Δ*hipA* background based on log$_2$ ratio vs sum of intensity, with the average median values from three replicates. The phosphorylation sites with log$_2$ ratio greater than 2 were more than four-fold up-regulated upon HipA$_{kp}$ overproduction compared to *K. pneumoniae* harboring the empty vector. **E and F)** The ratios for phosphoproteome were normalized to proteome and plotted here against the sum of intensity. **C)–F)** The phosphorylation sites that were more than four-fold up-regulated (black dots) are numbered and listed here. The size of each phosphorylation site in the scatter plot corresponds to its respective peptide score and scale is provided in the top left corner of the plot.

treatments of *K. pneumoniae* infections. We first determined the susceptibility of the *K. pneumoniae* isolate ATCC13883 to gentamicin, an aminoglycoside that inhibits protein synthesis, and found that a concentration above 3 μg/mL led to complete growth inhibition after 24 h of exposure. We then exposed the cells with and without $hipA_{kp}$ overexpression to 4 μg/mL gentamicin and did not observe any influence of $hipA_{kp}$ on the survival of *K. pneumoniae* cells (**Figs 4A and S4A**). These results were in agreement with a previously published antibiotic tolerance test where it was shown that $HipA_{ec}$ conferred protection against several different classes of antibiotics but not against the aminoglycoside tobramycin [13]. We next focused on ciprofloxacin, a fluoroquinolone antibiotic that inhibits DNA replication in growing cells. Ciprofloxacin was shown to be effective in the treatment of *K. pneumoniae* infections [26] and was previously used for testing antibiotic tolerance in *E. coli* [27]. We determined the susceptibility of the *K. pneumoniae* isolate ATCC13883 against ciprofloxacin and found that a concentration above 0.5 μg/mL led to complete growth inhibition after 24 h of exposure (**S4B Fig**). *K. pneumoniae* WT and Δ*hipA* expressing $hipA_{kp}$ from pBAD33 were exposed to 1 μg/mL ciprofloxacin for 2 h. Bacteria under uninduced conditions and bacteria harboring the empty pBAD33 vector were used as negative controls. The proteome measurement confirmed the overproduction of $HipA_{kp}$ in the induced strains after 2 h of ciprofloxacin treatment. Importantly, only cells expressing $hipA_{kp}$ showed survival after 2 h of ciprofloxacin treatment with a mean $\log_{10}$ CFU/ml value of 8. In contrast, the viability of the negative controls was six orders of magnitude lower ($\log_{10}$ CFU/ml value of 2) (**Fig 4B**). To further support the finding that $HipA_{kp}$ has kinase activity, *K. pneumoniae* was transformed with pBAD33 harboring a $hipA_{kp}$ gene that carries a mutation that leads to a D309Q amino acid substitution. The aspartate at position 309 is a catalytic residue in *E. coli* and the D309Q modification is expected to decrease kinase activity and ciprofloxacin tolerance in *K. pneumoniae*. Indeed, overproduction of the mutated $HipA_{kp}$ protein reduced the number of viable *K. pneumoniae* following ciprofloxacin treatment as compared with bacteria expressing the wild-type $hipA_{kp}$ allele (**S4C and S4D Fig**). We conclude from the data that $HipA_{kp}$ kinase activity contributes to antibiotic tolerance against ciprofloxacin in *K. pneumoniae* under the tested conditions, as also previously observed for $hipA_{ec}$ in *E. coli* [13].

## 1.5 Phosphoproteome analysis reveals potential $HipA_{kp}$ substrates in ciprofloxacin-treated *K. pneumoniae*

In order to detect the $HipA_{kp}$ targets potentially involved in the antibiotic survival, we overexpressed $hipA_{kp}$ in Δ*hipA K. pneumoniae*, treated the culture with ciprofloxacin for 2 h and compared the phosphoproteome results with uninduced and empty vector controls. At the proteome level, we identified a total of 1,889 proteins and confirmed $HipA_{kp}$ overproduction upon induction based on the volcano plot showing differential protein expression and correlation plot (**Figs 4C and S4E and Tab E in S1 Dataset**).

At the phosphoproteome level, we identified 547 phosphorylation sites that showed a high correlation between biological replicates (Pearson's correlation coefficient 0.88) (**S4F Fig and Tab F in S1 Dataset**). These phosphorylation sites were filtered for localization probability of >0.75. Following this, many phosphorylation sites were found to be up-regulated following ciprofloxacin treatment in $hipA_{kp}$ overexpressing cells, with approximately 20 phosphorylation sites consistently observed as up-regulated in both unnormalized and normalized phosphoproteome datasets (**Figs 4D, 4E, and S4G and Tab E in S1 Dataset**). This included the autophosphorylation sites and GltX phosphorylation at S239, as compared to untreated $hipA_{kp}$ overexpressing cells. The similarity in up-regulated phosphorylation sites with and without ciprofloxacin treatment indicates that overproduction of $HipA_{kp}$ induced tolerance to

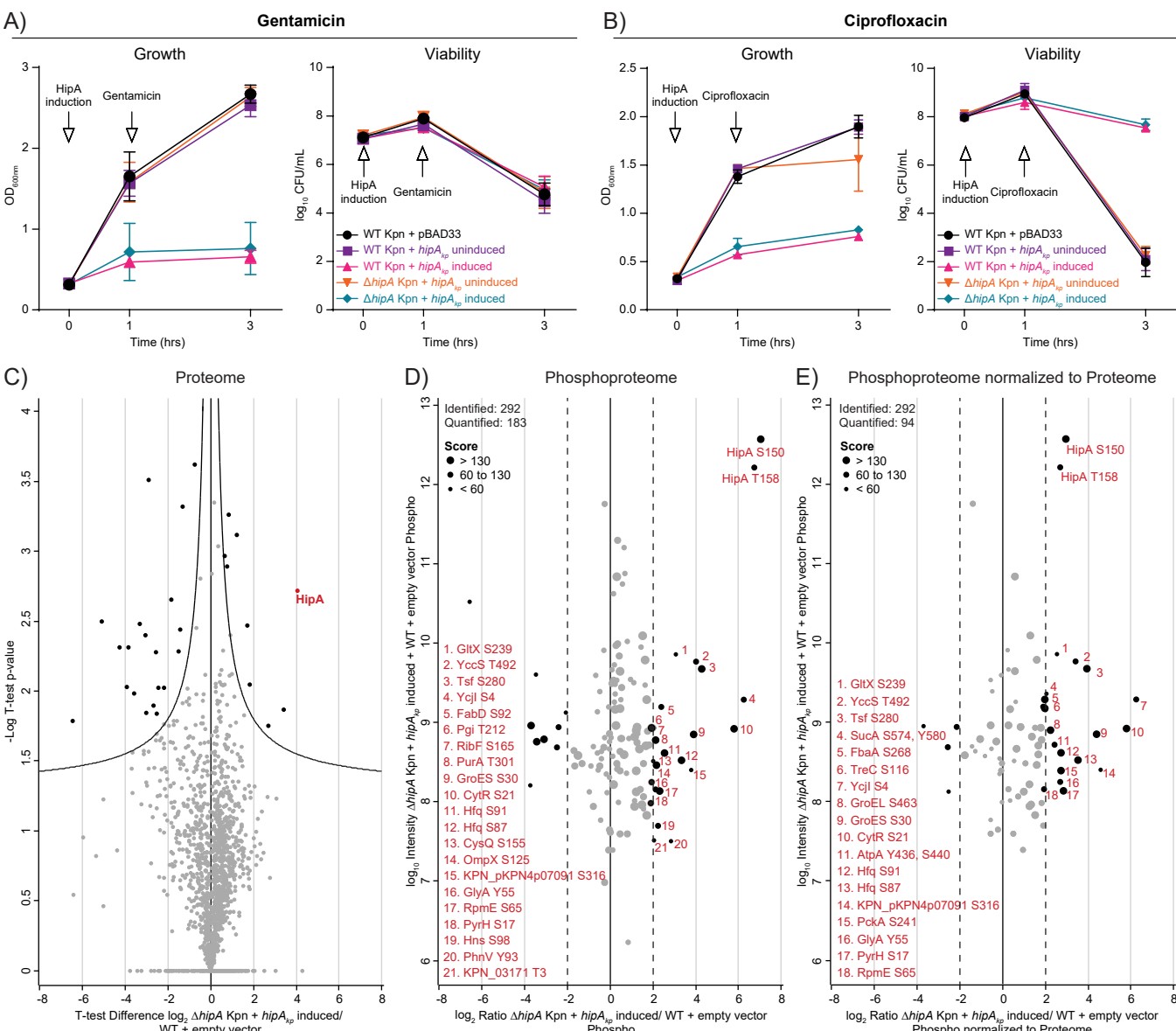

**Fig 4. Effect of overexpression of $hipA_{kp}$ on antibiotic tolerance in *K. pneumoniae*. A)** Growth of WT and *ΔhipA K. pneumoniae* strains transformed with pBAD33 and pBAD33::$hipA_{kp}$, respectively in which $hipA_{kp}$ expression was driven by the arabinose-inducible promoter. Strains were grown in LB Lennox medium and expression was induced at $OD_{600nm}$ of 0.3 for 1 h followed by treatment with 4 µg/mL gentamicin for 2 h. Growth was followed via optical density and viability by measuring colony-forming units. **B)** Growth of *K. pneumoniae* WT and *ΔhipA* strains transformed with pBAD33 and pBAD33::$hipA_{kp}$ with $hipA_{kp}$ expression driven by the arabinose-inducible promoter. Strains were grown in LB Lennox medium and expression was induced at $OD_{600nm}$ of 0.3 for 1 h followed by treatment with 1 µg/mL ciprofloxacin for 2 h. Growth was followed via optical density and viability by measuring colony-forming units. A) and B) Data shown are mean values ± SD of three biological replicates. **C)** Volcano plots showing differential protein expression to compare the proteome of ciprofloxacin-treated *ΔhipA K. pneumoniae* cells overproducing HipA$_{kp}$ with bacteria containing an empty vector. Student's one sample t-test was performed for n = 2 biological replicates. The x-axis represents $\log_2$ fold changes based on the ratio and the y-axis represents negative $\log_{10}$ of the Benjamini-Hochberg corrected (FDR<0.05) t-test p-value. The curves indicate confidence levels with p-value = 0.05. Proteins above the curve line on the left side are significantly up-regulated. **D and E)** Scatter plot showing the distribution of quantified phosphorylation sites after 2 h of ciprofloxacin treatment upon $hipA_{kp}$ overexpression, based on $\log_2$ ratio of *ΔhipA* + pBAD33::$hipA_{kp}$ to WT + empty vector for phospho unnormalized **(D)** and normalized to proteome **(E)** on the x-axis and $\log_{10}$ of sum of intensity on y-axis, with the average median values from two replicates. The phosphorylation sites with $\log_2$ ratio greater than 2 were more than four-fold up-regulated upon HipA$_{kp}$ overproduction compared to the empty vector sample. These phosphorylation sites (black dots) were numbered and listed on the right side of the plot in red. The size of each phosphorylation site in the scatter plot corresponds to its respective peptide score and scale is provided in the top left corner of the plot.

ciprofloxacin by phosphorylating target proteins. All the putative phosphorylation sites of HipA$_{kp}$ have been listed in **S4 Table** with the frequency of their occurrence to be more than four-fold increased or only identified upon *hipA$_{kp}$* overexpression (**Tabs D, F, and H in S1 Dataset**).

### 1.6 *K. pneumoniae* phosphoproteome reveals numerous pathways that are potentially regulated at the post-translational level

Our study provided the most comprehensive phosphoproteome dataset for *K. pneumoniae* so far [28–31], containing a total of 1439 phosphorylation sites from 741 proteins and a total of 2128 proteins from all experiments (**S3 and S4 Tables and Tabs I and J in S1 Dataset**). 37% phosphoproteins and 27% phosphopeptides occurred in multiple sets of independent experiments (**Fig 5A and 5B**). Further analysis revealed the distribution of phosphorylated serine, threonine, and tyrosine was 46.2%, 39.7%, and 14.0%, respectively (**Fig 5C**) Functional enrichment analysis performed on all identified phosphoproteins revealed the cellular processes potentially regulated by phosphorylation (**Fig 5D and Tab K in S1 Dataset**). Phosphoproteins were distributed across numerous cellular functions, with a significant proportion implicated in translation and RNA-binding. Additionally, many were associated with glycolysis, purine and pyrimidine biosynthesis, and DNA repair. The comparison of all previously published *K. pneumoniae* phosphoproteomics datasets [28–30] with our combined dataset (**Tab I in S1 Dataset**) revealed a large number of novel phosphoproteins and phosphopeptides that were previously unreported (**Fig 5E and 5F**). This dataset will serve as a valuable resource for researchers interested in studying protein phosphorylation in *K. pneumoniae*.

## Discussion

*K. pneumoniae* is a leading cause of nosocomial infections able to cause invasive infections and outbreaks in hospitals [32–34]. A particular threat to human health is the emergence of carbapenem-resistant strains that are associated with a high mortality rate and limited therapeutic options [35]. Moreover, there are increasing reports of carbapenem-resistant hypervirulent strains of *K. pneumoniae* [36–38]. Protein post-translational modifications (PTMs) play a vital role in regulating the function of various cellular processes, which can either lead to the activation or inactivation of the protein activity [39]. Protein phosphorylation is one of the major PTMs that provides a universal mechanism to regulate a large variety of processes and several recent quantitative phosphoproteomics studies have focused on the molecular function of Ser/Thr kinases, such as HipA, HipA7, and HipH (YjjJ) in *E. coli* [19,40]. Due to the limited number of studies addressing antibiotic tolerance of *K. pneumoniae* at the molecular level, we investigated the phosphoproteome of *K. pneumoniae* cells after the overproduction of the kinase HipA. We hypothesized that the *hipB/A* operon from *K. pneumoniae* (*hipB/A$_{kp}$*) has similar functions to the well-characterized *hipB/A* operon from *E. coli* (*hipB/A$_{ec}$*), which is implicated in antibiotic tolerance and persistence. Although the primary sequence identity was lower than 70%, structural regions and residues essential for HipA kinase activity were conserved in both organisms. In addition, the alignment of predicted and experimentally determined 3D structures revealed a high level of conservation of the overall structure with a low RMSD value of 0.4 Å [41].

Using MS-based proteomics, we first showed that overexpression of *hipA$_{kp}$* in *E. coli* led to the inhibition of growth, which could be restored upon simultaneous *hipB$_{kp}$* induction. These experiments indicated the *in vivo* kinase activity of the HipA$_{kp}$ and its interplay with HipB$_{kp}$. Phosphoproteome analysis upon *hipA$_{kp}$* induction in *E. coli* showed a variety of potential substrates of HipA$_{kp}$ including the well-known target of *hipA$_{ec}$*, GltX at S239. Since *K. pneumoniae*

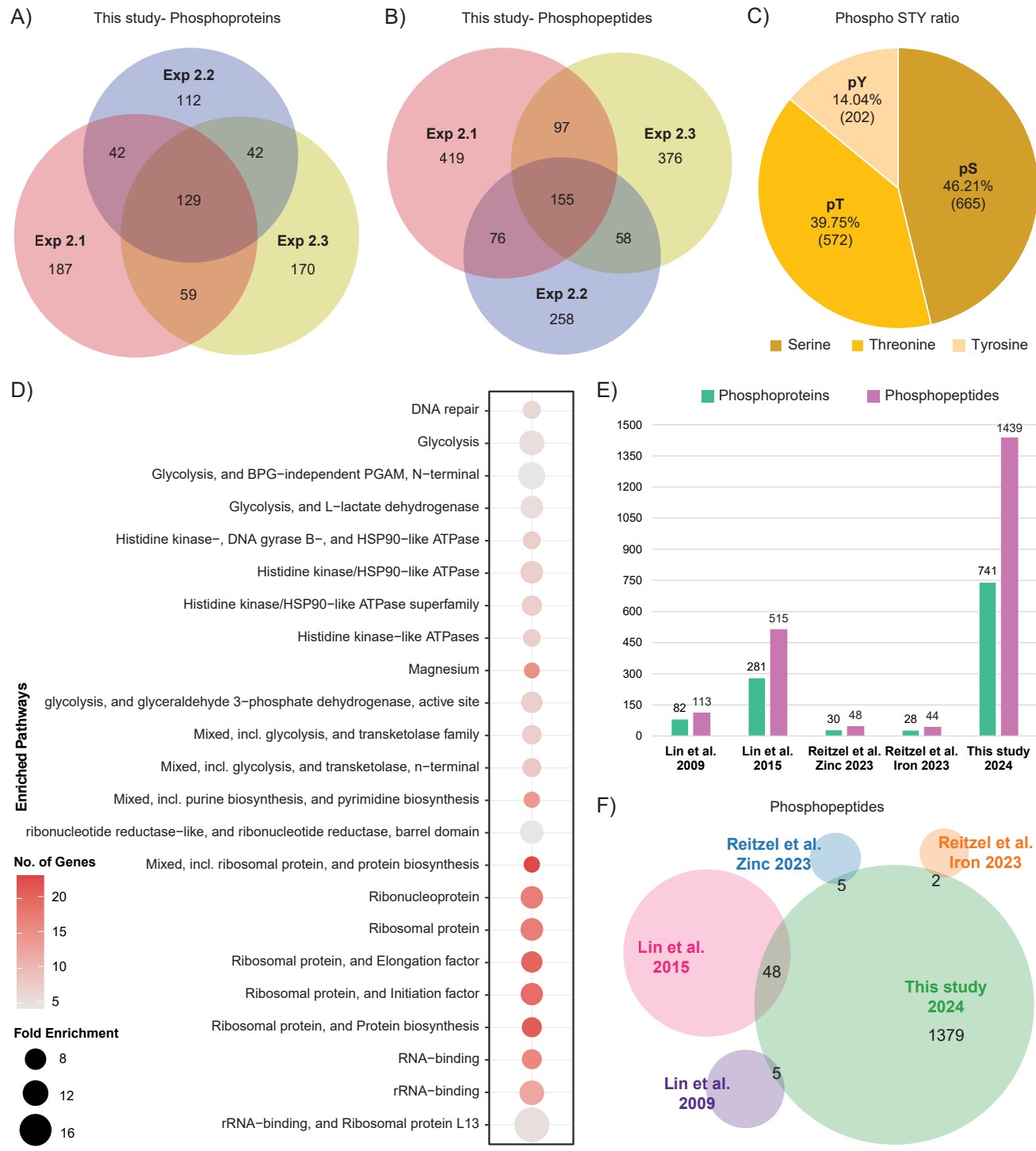

**Fig 5. Overview of the *K. pneumoniae* phosphoproteome datasets.** A) and B) Venn diagram comparing the three phosphoproteome datasets of *K. pneumoniae* obtained in this study showing the overlap between quantified phosphoproteins (**A**) and phosphopeptides (**B**). C) Distribution of all phosphorylated serine (pS), threonine (pT) and tyrosine (pY) identified in the combined dataset from three experiments, showing the number of phosphorylation sites of each amino acid in bracket. **D)** Functional enrichment analysis of phosphorylated proteins showing the fold enrichment and number of proteins phosphorylated in the indicated pathway. **E)** Comparison of all previously published phosphoproteome datasets for *K. pneumoniae* with our dataset for the number of phosphoproteins and phosphopeptides identified. **F)** Analysis of the overlap between the number of phosphopeptides identified in *K. pneumoniae* in our study and those in published *K. pneumoniae* phosphoproteome studies [28–31].

and *E. coli* have different genetic backgrounds, we expected different HipA substrate pools in the two organisms. Therefore, we next analyzed HipA$_{kp}$ activity in *K. pneumoniae*. Upon deletion of *hipA$_{kp}$* in *K. pneumoniae* (Δ*hipA)* we observed no significant difference in growth and viability in comparison with the WT, indicating that *hipA$_{kp}$* is not an essential gene in the exponential phase, although we observed some reduction in viability in the late stationary phase which might suggest its role in stationary phase or non-growing cells. We performed further experiments in WT and Δ*hipA* background to rule out the possibility of any potential effect of the native *hipA* gene on the phosphoproteome. As expected, overexpression of *hipA$_{kp}$* in both WT and Δ*hipA Klebsiella* background was toxic to the cells and this could be partially rescued with *hipB$_{kp}$* overexpression. Overexpression of a kinase-dead mutant of *hipA$_{kp}$* D309Q showed that the kinase activity of HipA is required for the toxic phenotype and antibiotic tolerance as overexpression of the mutant *hipA* gene reduced tolerance against ciprofloxacin. However, additional experiments are needed for a more comprehensive analysis of HipA activity as a kinase. In *E. coli* with *hipA$_{kp}$* overexpression, we observed almost complete complementation with overproduction of HipB$_{kp}$ but in *Klebsiella*, the complementation was only partial. The reason for this is at present unclear, however, we note that overexpression of *hipA$_{kp}$* and *hipB$_{kp}$* in *E. coli* points to a canonical toxin/antitoxin pair.

The normalization of phosphoproteome data to the proteome resulted in a loss of approximately half of the phosphorylation sites, likely due to the absence of corresponding protein ratios for these sites. Despite this reduction, most up-regulated phosphorylation sites remained largely unaffected by normalization, with the notable exception of HipA. Due to HipA over-production, normalization with proteome reduced the high levels of HipA autophosphorylation sites, bringing them down to a slight up-regulation level in comparison to the unnormalized data. Phosphoproteomics analysis upon *hipA$_{kp}$* induction in *K. pneumoniae* showed a total of 63 common phosphoproteins between *E. coli* and *K. pneumoniae* datasets. Among them was GltX, which appears to be a prominent target in both organisms. Several additional proteins were detected as potential targets of HipA$_{kp}$, most of them associated with essential biological processes. In our *K. pneumoniae* phosphoproteomics datasets, we observed 34 putative HipA$_{kp}$ substrates to be either exclusively phosphorylated upon *hipA$_{kp}$* overexpression or more than four-fold increased in this condition, including *hipA*, *gltX*, *rcsB*, *tsf*, *ybiT* and *ycjI* (**S5 Table**). These sites should be prioritized as the most likely HipA$_{kp}$ targets for future biological follow-up experiments. A majority of these substrates are phosphorylated at either N- or C-termini in the regions that bind DNA or RNA or other proteins, hinting towards the function of these phosphorylation events in modulating processes dependent on protein/protein or protein/nucleic acid interactions (**S5 Fig**). For example, SeqA phosphorylation at S36 and S46 occurs in the N-terminal region, essential for self-association [42], Tig phosphorylation at S4 is present in the N-terminal ribosome-binding region [43] and CueR phosphorylation at S4 is located in the N-terminal DNA-binding region of the protein [44]. Conversely, proteins RbfA, BipA, FtsK, Hns, Rne, RpsA, and Tsf are phosphorylated close to their C-termini. RbfA phosphorylation at S110 is in the region required for stable 30S ribosomal association *in vitro* and efficient 16S rRNA processing [45]. BipA phosphorylation at S490 is in the region that binds to the A-site of tRNA in *E. coli* [46,47], FtsK phosphorylation at S1373 is in the region essentially involved in chromosome segregation [48], whereas Hns at S98 in the DNA-binding region [49].

Interestingly, several of the observed phosphoproteins were previously reported to be phosphorylated or involved in phage infection. The ribonuclease E Rne, is a component of the RNA degradosome interacting with proteins such as RhlB helicase, enolase and PNPase via its C-terminal RNA-binding domain [50]. Rne has been shown to be phosphorylated by the T7 phage kinase gp-0.7 (PK) to inhibit its activity upon phage infection [51]. Similarly, small

ribosomal subunit protein S1 RpsA is also known to be phosphorylated by PK to enhance phage protein production [52]. The elongation factor-Ts (Tsf), along with elongation factor-Tu (Tuf) and RpsA, form a complex and are part of the host-provided phage Qbeta RNA polymerase complex [53]. The phosphorylation of a similar set of proteins by *hipA* and T7 phage kinase 0.7 suggests a potential link between these kinases, requiring further investigation.

The role of wild-type and kinase-dead variants of HipA$_{ec}$ in tolerance has been studied against many antibiotics, resulting in no or limited tolerance in the absence of HipA activity against ofloxacin, mitomycin C, and cefotaxime but not against tobramycin [13]. Antibiotic tolerance observed in the *K. pneumoniae* strains overexpressing *hipA$_{kp}$* seemed to be also specific to certain classes of antibiotics. We observed that *K. pneumoniae* cells overexpressing *hipA$_{kp}$* showed increased survival after treatment with ciprofloxacin, but not against gentamicin. Although this is in agreement with earlier results from *hipA$_{ec}$* [13], the reason for this is currently unclear. It is conceivable that cell growth inhibition due to overproduced HipA$_{kp}$ can provide survival benefits during treatment with ciprofloxacin, which acts on dividing cells and inhibits their cell cycle. Gentamicin, on the other hand, inhibits protein synthesis by binding to 30S ribosomes. As already reported for *hipA$_{ec}$*, persistent cells have a basal level of protein synthesis which is potentially essential for their survival [27]. Therefore, complete inhibition of the protein synthesis by gentamicin can lead to the killing of the cells expressing *hipA$_{kp}$*. However, further experiments are needed to address this. More than half of the phosphorylation sites obtained in ciprofloxacin-treated *hipA$_{kp}$*-overexpressing *K. pneumoniae* cells were identical with the phosphorylation sites in *hipA$_{kp}$*-overexpressing *K. pneumoniae* cells in the absence of antibiotic. This revealed that the tolerance mechanism is likely based on GltX-mediated activation of the stringent response, although many of the above-mentioned substrates also act on other levels of cellular processes to ensure survival.

Finally, our study provides a basis for the comparison of the molecular phenotype of the ciprofloxacin treatment in *E. coli* and *K. pneumoniae*. Among 821 phosphoproteins detected in a recent study of ciprofloxacin-treated *E. coli* cells [54], 100 phosphoproteins were also detected in our ciprofloxacin-treated *hipA$_{kp}$*-overexpressing *K. pneumoniae* cells. This indicates that numerous proteins are phosphorylated in both organisms in response to ciprofloxacin treatment. Autophosphorylation on HipA$_{ec}$ at S150 was up-regulated during ciprofloxacin treatment in *E. coli cells* even without any overproduction of HipA$_{ec}$ indicating that ciprofloxacin treatment promotes the activity of this protein in *E. coli*. This also implies that this site may have a role in the regulation of kinase activity during antibiotic treatment [54]. We directly compared our phosphoproteome dataset with previously reported phosphorylation sites for *K. pneumoniae* before any statistical testing or filtering for localization probability. We noted that the two studies by Li *et al.* used a different strain of *K. pneumoniae* for the experiment and also different software for data processing and analysis, which may have resulted in some differences in the protein names and position of phosphorylation for the phosphoproteomics data and thereby resulted in lower overlap. However, despite these and probably other differences (growth conditions, etc.) between the previously published studies and our study, we think it is important and insightful to compare all the data.

Combined, our results provide a rare insight into molecular mechanisms of post-translational regulation of antibiotic tolerance in *K. pneumoniae* and provide a resource for further studies on this important human pathogen.

## Materials and methods

### 1. Bacterial strains and plasmids

All strains, primers and plasmids used in this work are listed in the table below (**S1 and S2 Tables**). Due to the high homology of *hipA$_{kp}$* with *hipA$_{ec}$*, the *hipA$_{kp}$* gene was cloned with the

same Shine-Dalgarno sequence containing the alternative starting codon GTG as used in previous work [19]. *hipA*$_{kp}$ and *hipB*$_{kp}$ genes were amplified from the *Klebsiella pneumoniae* subsp. *pneumoniae* reference strain ATCC13883 with primers HipA.kpn.SD8.GTG.pBAD33-for and HipA.kpn.pBAD33-rev, and hipBkp.pGOOD-for and hipBkp.pGOOD-rev, respectively. The pBAD33 vector and pGOOD vector were digested with *Xba*I and *Bgl*II restriction enzymes, respectively and finally pBAD33 was ligated with *hipA*$_{kp}$ and pGOOD with *hipB*$_{kp}$ amplified PCR product using Gibson assembly mix [55]. The resulting plasmids were transformed into Top10 *E. coli* competent cells and confirmed by PCR for the presence of the gene of interest and sequenced before transforming them into final working strains in both *E. coli* and *K. pneumoniae*. The Q5 Site-Directed Mutagenesis Kit (NEB) was used to introduce a GAC to CAG codon exchange into *hipA*$_{kp}$ on pBAD33 leading to HipA$_{kpD309Q}$ according to the manufacturer's instructions. The plasmid was sequenced and cloned into *K. pneumoniae*.

### Generation of *K. pneumoniae* Δ*hipA* mutant

A markerless in-frame deletion of *hipA* was generated by amplifying approximately 600 bp up- and down-stream of *hipA*. PCR products were fused with SOE-PCR and cloned with In-Fusion cloning (TakaraBio) into the suicide vector pKNOCK-Km, a gift from Mikhail Alexeyev (Addgene #46262). The *sacB* gene including its promoter was amplified from the plasmid pEXG2 and cloned into pKNOCK-Km to introduce a counter selection marker. Plasmid inserts were verified by Sanger sequencing. The resulting plasmids were transferred into *E. coli* S17λ*pir* for the transformation of *K. pneumoniae* ATCC13883 by conjugation. *K. pneumoniae* merodiploids were selected on LB agar plates supplemented with 100 μg/ml kanamycin, streaked for single colony isolation and then incubated in LB without selection overnight. Counter selection was performed with 15% sucrose, and kanamycin sensitive colonies were tested for the deletion of *hipA* by PCR.

### 2. Bioinformatic analysis

The sequence of HipA$_{kp}$ from *Klebsiella pneumoniae* subsp. *pneumoniae* ATCC13883 was analyzed for its identity within *Klebsiella* genus and other organisms using protein BLAST at the NCBI server (https://blast.ncbi.nlm.nih.gov/Blast.cgi). For the initial pBLAST search, the search was limited to Klebsiella (taxid:570) as the organism and 1,000 maximum target sequences in the algorithm parameters. Percent identity and number of hits per species and subspecies were extracted from the pBLAST result. For the analysis of HipA$_{kp}$ identity across all organism without limiting the search to Klebsiella, we performed the pBLAST search for top 5,000 hits and plotted the results. Both the graphs were generated using the online tool, Shiny BoxPlotR (http://shiny.chemgrid.org/boxplotr/). The pairwise sequence alignment between HipA$_{kp}$ and HiA$_{ec}$, or HipB$_{kp}$ and HipB$_{ec}$ proteins was performed with EMBOSS Needle [56] using the default settings.

### 3. Growth experiments

**Growth of WT *E. coli* cells overexpressing *hipA*$_{kp}$.** Overnight pre-cultures of WT *E. coli* MG1655, containing pBAD33 empty vector and pBAD33::*hipA*$_{kp}$, were prepared in liquid medium (Luria-Bertani medium from Roth) supplemented with 0.4% (w/v) glucose, 25 μg/mL chloramphenicol for the maintenance of pBAD33 plasmid. The next day cultures were started at an OD$_{600nm}$ of 0.08 and induced with 0.2% arabinose for pBAD33 plasmid upon reaching an OD$_{600}$ of 0.3 for 1 h and harvested afterwards for (phospho)proteome analysis. The OD$_{600nm}$ and number of CFU were measured and calculated before and after 1 h of induction.

The experiment was performed in 3 biological replicates and results were visualized using GraphPad Prism 8.0.1.

**Growth of WT *E. coli* cells overexpressing $hipA_{kp}$, and $hipB_{kp}$.** Overnight pre-cultures of WT *E. coli* MG1655, containing only pBAD33::$hipA_{kp}$, and both pBAD33::$hipA_{kp}$ + pGOOD::$hipB_{kp}$ were prepared in liquid medium (Luria-Bertani medium from Roth) supplemented with 0.4% (w/v) glucose along with 25 µg/mL chloramphenicol and 10 µg/ml tetracycline for the maintenance of pBAD33 and pGOOD plasmid, respectively. The following day, cultures were initiated at a starting $OD_{600nm}$ of 0.08 with the control samples left uninduced and the others were induced with 0.2% arabinose for pBAD33 plasmid and 1 mM IPTG for pGOOD plasmid in a 24-well plate (Greiner) and incubated at 37°C and 300 rpm in a plate reader (Tecan). Three biological replicates were performed.

**Growth and survival of *K. pneumoniae* Δ*hipA*.** LB Lennox was inoculated to an $OD_{600nm}$ of 0.05 with overnight cultures of *K. pneumoniae* wild type and Δ*hipA*. A 1 mL aliquot of the cultures was transferred to a 24-well plate (Greiner) and incubated at 37°C and 300 rpm in a plate reader (Tecan). The $OD_{600nm}$ was measured every 30 mins for 24 h with four reads per well and the medium blank value was subtracted from the experimental values. Three independent experiments in triplicate were performed.

**Growth and survival of *K. pneumoniae* overexpressing *hipA*.** LB Lennox supplemented with 0.4% glucose and 50 µg/ml chloramphenicol was inoculated with *K. pneumoniae* wild-type pBAD33, wild-type pBAD33::$hipA_{kp}$ and Δ*hipA* pBAD33::$hipA_{kp}$ and incubated overnight. The next day, 250 ml LB Lennox was inoculated for each strain to an $OD_{600nm}$ of 0.05 and incubated until an $OD_{600nm}$ of approximately 0.3 was reached. Aliquots were taken to determine the number of CFU/mL by plating serial dilutions. Each culture was split into two. One fraction was induced with 0.2% arabinose and the other one was left uninduced. At 1 and 2 h post-induction, the $OD_{600nm}$ and CFU/mL was determined. Three independent experiments were performed.

**Growth of *K. pneumoniae* overexpressing *hipA* and *hipB*.** *K. pneumoniae* Δ*hipA* harboring pBAD33::$hipA_{kp}$ alone or together with pGOOD::$hipB_{kp}$ were grown overnight in LB Lennox supplemented with 0.4% glucose and where necessary with 50 µg/ml chloramphenicol and 10 µg/ml tetracycline. The bacteria were used to start cultures at an $OD_{600nm}$ of 0.05 in LB Lennox without supplements and grown to early exponential phase. The $OD_{600nm}$ was adjusted to 0.3 and two fractions per strain were prepared, one for inducing conditions and the other one was left uninduced. Expression of $hipA_{kp}$ and $hipB_{kp}$ was induced with 0.2% arabinose and 1 mM IPTG, respectively. 1 ml of each culture was added to a 24-well plate (Greiner) and the $OD_{600}$ was recorded every 15 mins for 5.5 h at 37°C and 300 rpm in a plate reader (Tecan). The average blank absorbance was subtracted from the sample values. Three independent experiments were performed in triplicate.

## 4. Antibiotic tolerance test of *K. pneumoniae* overexpressing $hipA_{kp}$

**Gentamicin and ciprofloxacin sensitivity of *K. pneumoniae* ATCC13883.** Antibiotic resistance of the wild-type *K. pneumoniae* ATCC13883 was tested by inoculating 2 mL LB Lennox supplemented with gentamicin (Sigma Aldrich) at concentrations ranging from 0 to 4 µg/ml and ciprofloxacin (Sigma Aldrich) at concentrations ranging from 0 to 5 µg/ml to an $OD_{600nm}$ of 0.05. The cultures were incubated at 200 rpm and 37°C for 24 h and the $OD_{600nm}$ was measured.

**Gentamicin and ciprofloxacin tolerance test of *K. pneumoniae* with $hipA_{kp}$ overexpression.** *K. pneumoniae* wild-type pBAD33, WT pBAD33::$hipA_{kp}$ and Δ*hipA* pBAD33::$hipA_{kp}$ were grown overnight in LB Lennox supplemented with 0.4% glucose and 50 µg/ml

chloramphenicol. 200 mL LB Lennox was inoculated with the overnight cultures to an initial $OD_{600nm}$ of 0.05 and incubated until an $OD_{600nm}$ of approximately 0.3 was reached. Each culture was split into two subcultures, one was treated with 0.2% arabinose and the other one was left untreated. Following arabinose induction for 1 h, 4 µg/ml of gentamicin or 1 µg/ml ciprofloxacin was added. Immediately before induction, at 1 h post-induction and 2 h post-antibiotic treatment, the absorbance and colony forming units were determined. Three independent experiments were performed. Graphs were generated using GraphPad Prism 8.0.1.

## 5. Cell lysis and protein precipitation

Harvested cells were centrifuged at 4,000 *g* and the pellet was stored at -80˚C. The pellet was then resuspended in an SDS lysis buffer containing 40 mg/ml SDS (sodium dodecyl sulfate), 100 mM Tris-HCl pH 8.6, 10 mM EDTA, 5 mM glycerol-2-phosphate, 5 mM sodium fluoride, 1 mM sodium orthovanadate and 1 tablet of complete protease inhibitors (Roche). The cell lysate was sonicated at 40% amplitude for 30 secs cycle at least five times or until a transparent, non-viscous lysate was obtained. The cell debris was pelleted by centrifugation at 13,000 *g* for 30 mins and the supernatant was collected for protein precipitation using methanol and chloroform method. The obtained protein pellet was air-dried and dissolved in denaturation buffer (6 M urea, 2 M thiourea and 10 mM Tris pH 8.0). The protein concentration was determined by using standard Bradford assay (Bio-Rad).

## 6. Protein in-solution digestion

For each phosphoproteomics experiment, 3–6 mg protein per sample (strain/condition) was used. Briefly, precipitated proteins were reduced by using 1 mM dithiothreitol (DTT) for 1 h and then alkylated using 5.5 mM iodoacetamide (IAA) for an additional 1 h in the dark with constant shaking at 700 rpm. Half of the protein from each sample was diluted with four times volume of 62.5 mM Tris pH 8.0 and 12.5 mM $CaCl_2$ and digested with the enzyme chymotrypsin (1:100 w/w) overnight at room temperature (RT) in a shaker. The other half of the protein was pre-digested with the endoproteinase LysC (1:100 w/w) for 3 h and then diluted with four times volume of milli-Q water, adjusted to a pH higher than 8.0 and supplemented with the enzyme LysC (1:100 w/w) (for *E. coli* samples) and trypsin (1:100 w/w) (for *K. pneumoniae* samples) for overnight digestion at RT and shaking. The reaction was then stopped by acidification with trifluoroacetic acid (TFA) to pH 2.0 and centrifuged to get rid of precipitates.

## 7. Solid phase extraction and dimethyl labeling

Acidified peptides were then purified by solid phase extraction on Sep-Pak C18 cartridges (Waters, Milford, MA) and labeled using triplex stable isotope dimethyl labeling as previously described [57]. Briefly, C18 columns were activated with methanol and equilibrated with Solvent A* [2% (v/v) acetonitrile (AcN) and 1% (v/v) formic acid (FA)]. The digested and acidified peptide samples were loaded and later, the column was washed with HPLC Solvent A [0.1% (v/v) FA]. These samples were then labeled with 2.5 ml of the respective labeling solutions: $CH_2O$ (Sigma-Aldrich) and $NaBH_3CN$ (Fluka) for Light label, and $CD_2O$ (Sigma-Aldrich) and $NaBH_3CN$ for Medium label, and $C_{13}D_2O$ (Sigma-Aldrich) and $NaBD_3CN$ (96% D, Isotec) for Heavy label. The labeling solutions were flushed with approximately 10–15 mins contact time through with the column. Labeled peptides were washed again with HPLC Solvent A on the column and eluted with 600 µl HPLC Solvent B [80% (v/v) AcN in 0.5% (v/v) FA].

## 8. Labeling efficiency and mixing check

For the validation of labeling efficiency and accurate mixing of the labeled peptides, two sets of 5 µg of each eluted labeled sample were used for LC-MS/MS measurements, separately (for labeling efficiency) and mixed in 1:1:1 ratio for a pilot mixing check measurement. Adjustments were made based on the mixing ratios to achieve a target ratio close to 1 to ensure optimal quantification. The labeling efficiency, for all labels, was $>= 95\%$.

## 9. Phosphopeptide enrichment

Phosphopeptides were enriched by titanium dioxide ($TiO_2$) beads with a ratio of 1:10 (beads: protein ratio) for five consecutive rounds of enrichment for 10 mins each. After mixing the labeled samples together and taking out an aliquot of 10 µg for proteome analysis, the peptides were acidified. $TiO_2$ beads were washed with 80% (v/v) AcN and 6% (v/v) TFA and incubated with the samples with constant mixing. The beads with bound phosphopeptides were washed again to remove any unbound or acidified peptides using 80% AcN and 6% TFA. These beads were then loaded onto C8 (Empore) StageTips and further washed with a wash buffer [80% AcN and 1% TFA] and Solvent B [80% AcN and 0.1% FA]. Phosphopeptides were eluted with 30 µl of elution solution I [1.25% (v/v) ammonium hydroxide of pH>10.5 into a tube containing 20 µl of 20% (v/v) TFA. This was followed by 70 µl elution solution II [5% (v/v) ammonium hydroxide in 60% (v/v) AcN (pH>10.5)] and finally with 20 µl of elution solution III [60% (v/v) AcN, 1% (v/v) TFA]. Each elution solution took at least 15 mins, at 1000–1500 rpm to elute. Acetonitrile was evaporated from eluates by vacuum centrifugation and samples were acidified to pH 2.0, and purified by StageTips.

## 10. Peptide purification by StageTips

Before LC-MS/MS measurements, samples for proteome analysis and eluted phosphopeptides were desalted and purified on C18 StageTips [58]. Briefly, reverse-phase chromatography was applied using C18 discs (Empore). The discs were activated with methanol and equilibrated with Solvent A*. The acidified peptides were loaded onto the discs and washed with Solvent A. Peptides were eluted with 50 µl of Solvent B and vacuum centrifuged to evaporate AcN. The final sample volume was adjusted with Solvent A and final 10% (v/v) of Solvent A*.

## 11. LC-MS/MS measurement

Purified peptides were separated by an online coupled EASY-nLC 1200 system (Thermo Fischer Scientific) to an Orbitrap Exploris 480 spectrometer (Thermo Fischer Scientific) through a nano-electrospray ion source (Thermo Fischer Scientific). Chromatographic separation was performed on a 20 cm long and 75 µm inner diameter analytical column packed in-house with reversed-phase ReproSil-Pur C18-AQ 1.9 µm particles (Dr. Maisch GmbH). Peptides were loaded onto the column at 40°C, with 1 µl/min flow rate under a maximum back-pressure of 850 bar. The gradient was applied using HPLC Solvent A and 10 to 50% Solvent B at a 200 nl/min constant flow rate. Labeling efficiency samples were eluted using 36 mins, mixing check using 60 mins, phosphopeptides using 60 mins and proteome samples using 130 mins or 230 mins gradients. Mass spectrometer was operated in positive ion and data-dependent acquisition mode. The acquisition of all full MS was in the scan range 300–1750 m/z at a resolution of 60k. For proteome measurements, the 20 most intense peptides were picked for HCD fragmentation at 15k resolution and for phosphoproteome at 30k resolution. The normalized collision energy was set to 28% and dynamically excluded the mass of sequenced

precursors for 30 secs from repeated fragmentation. The ions with single, unassigned or charge higher than six were also excluded from selection for fragmentation.

## 12. MS data processing with MaxQuant

The acquired raw files were processed using the MaxQuant software (version 2.2.0.0) [59]. Raw files from each set of experiments were processed separately in a similar manner (**S3 Table**). In total, we used **130** files of which **110** were files from phosphopeptide enrichment fractions. The obtained peak list was searched using Andromeda search engine integrated in MaxQuant [60] against *E. coli* K-12 MG1655 proteome (Taxonomy ID 83333) (released 30.01.2024, 4416 entries), and *Klebsiella pneumoniae* subsp. *pneumoniae* MGH78578/ ATCC700721 (Taxonomy ID 272620) (released 10.11.2022, 5127 entries), and common potential contaminants list. All search parameters were kept to default except the ones mentioned here. Labeling was set to three multiplicity, with Light: DimethylLys0 and DimethylNter0, Medium: DimethylLys4 and DimethylNter4, and Heavy: DimethylLys8 and DimethylNter8. Phospho (STY) was added as a variable modification for phosphopeptide-enriched files. Proteome and phosphoproteome files were grouped separately to only look for Phospho (STY) modification in phospho-files. pHis and pAsp were not included as variable modifications, due to low pH conditions used during phosphoenrichment. For Lys-C digestion, Lys-C enzyme was selected with a maximum of two missed cleavages allowed, similarly for trypsin with two missed cleavages allowed, and for chymotrypsin five missed cleavages allowed. To increase the number of quantified features, "match between runs" was enabled. The "Re-quantify" option was also enabled to allow the quantification of dimethyl-labeled pairs. Different experiments were processed separately for individual analysis and also together for the final set of phosphoproteome data.

## 13. Data analysis with Perseus

For the statistical analysis of MaxQuant output data, we used Perseus software (version 1.6.5.0.) [61] and the figures were edited using Adobe Illustrator. All contaminants, reverse hits, and diagnostic peaks were filtered out from the Phospho (STY) table. Average median values of phosphorylation site and proteome ratios were $\log_2$ transformed and plotted against $\log_{10}$ transformed sum of the intensities of phosphopeptides or proteins. Phosphorylation data was also normalized to proteome by dividing phospho ratios to proteome and plotted against $\log_{10}$ of the sum of intensity. Significantly regulated sites were determined by applying a threshold of 2 on a $\log_2$ scale (four-fold). The correlation between the two replicates was plotted with the density estimation feature in Perseus and plotted $\log_2$ ratio of the two replicates and calculated the value of Pearson's correlation coefficient. Likewise, for the Protein groups files, contaminants, reverse hits and only identified by sites proteins were filtered. After the $\log_2$ transformation of ratios, density estimation was performed. Scatter plots were prepared for the reproducibility between the replicates and Pearson's correlation was calculated. Further statistical analysis was performed for all proteome data in Perseus. Student's one-sample t-test was performed to prepare volcano plots showing differential expression of protein and phosphorylation sites based on the T-test difference of $\log_2$ ratio on the x-axis and the negative $\log_{10}$ of the Benjamini-Hochberg corrected (FDR < 0.05) p-value on the y-axis. Functional enrichment analysis of the final phosphoproteome dataset was performed using the online tool ShinyGO 0.77 (http://bioinformatics.sdstate.edu/go/) [62]. All Protein IDs from the combined Phospho (STY) table (Tab I in **S1 Dataset**) were added to the list and the species was selected as "Klebsiella pneumoniae". Using default values with an FDR cut-off of 0.05, we performed the enrichment analysis and obtained a table containing results of all enriched pathways, 149

pathways in total (**Tab K in S1 Dataset**). The data was further filtered using MS Excel for the number of genes per pathway ≥4, fold enrichment ≥4 and enrichment FDR ≤0.01. This resulted in 23 pathways that were used for plotting the highly enriched phosphorylated proteins in our dataset (**Fig 5D**).

## Supporting information

**S1 Fig. Additional bioinformatic analysis of *hipBA* from *K. pneumoniae*.**
(PDF)

**S2 Fig. Analysis of the effect of *hipA*$_{kp}$ overexpression on proteome and phosphoproteome of in *E. coli*.**
(PDF)

**S3 Fig. Additional analysis of proteome and phosphoproteome data from *hipA*$_{kp}$ overexpression in *K. pneumoniae*.**
(PDF)

**S4 Fig. Additional analysis of proteome and phosphoproteome data from *hipA*$_{kp}$ overexpression in *K. pneumoniae* after antibiotic treatment.**
(PDF)

**S5 Fig. Localization and putative role of phosphorylation in potential substrates of HipA$_{kp}$.**
(PDF)

**S1 Table. Bacterial strains and plasmids.**
(DOCX)

**S2 Table. DNA oligonucleotides.**
(DOCX)

**S3 Table. Overview of experiments for LC-MS/MS measurements.**
(DOCX)

**S4 Table. Analysis of proteome and phosphoproteome data from *Klebsiella pneumoniae*.**
(DOCX)

**S5 Table. List of putative substrates of HipA$_{kp}$.**
(DOCX)

**S1 Dataset. Protein groups and phosphorylation sites identified in this study.**
(XLSX)

## Acknowledgments

We thank Fabio Lino Gratani and Claudia Cavarischia-Rega for the initial inputs for the project, and Libera Lo Presti for critical reading of the manuscript and valuable comments. P.N. is a member of the International Max Planck Research School 'From Molecules to Organisms'.

## Author Contributions

**Conceptualization:** Payal Nashier, Sandra Schwarz, Boris Macek.

**Data curation:** Payal Nashier, Isabell Samp, Marvin Adler, Fiona Ebner, Lisa Thai Lê, Marc Göppel, Sandra Schwarz.

**Formal analysis:** Payal Nashier, Sandra Schwarz, Boris Macek.

**Funding acquisition:** Ivan Mijakovic, Sandra Schwarz, Boris Macek.

**Investigation:** Payal Nashier, Sandra Schwarz, Boris Macek.

**Methodology:** Payal Nashier, Isabell Samp, Marvin Adler, Fiona Ebner, Lisa Thai Lê, Marc Göppel, Carsten Jers, Sandra Schwarz, Boris Macek.

**Project administration:** Boris Macek.

**Resources:** Ivan Mijakovic, Sandra Schwarz, Boris Macek.

**Software:** Boris Macek.

**Supervision:** Ivan Mijakovic, Boris Macek.

**Validation:** Payal Nashier, Boris Macek.

**Visualization:** Payal Nashier.

**Writing – original draft:** Payal Nashier, Sandra Schwarz, Boris Macek.

**Writing – review & editing:** Payal Nashier, Carsten Jers, Ivan Mijakovic, Sandra Schwarz, Boris Macek.

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
