## [Decision Letter · Decision Letter 0]

29 Aug 2024

Dear Prof. Macek,

Thank you very much for submitting your manuscript "Deep phosphoproteomics of Klebsiella pneumoniae reveals HipA-mediated tolerance to ciprofloxacin" for consideration at PLOS Pathogens. Your manuscript was reviewed by members of the editorial board and by four independent reviewers. In light of the reviews (below), we would like to invite the resubmission of a substantially revised version that takes into account the reviewers' comments. Please address all the concerns of the reviewers; specifically, you must provide missing information on the statistical analyses as described by Reviewer 4, assay the function of HipA with a kinase mutant as noted by Reviewers 2 and 3, and examine HipA production in antibiotic resistant strains also as mentioned by Reviewers 2 and 3.

We cannot make any decision about publication until we have seen the revised manuscript and your response to the reviewers' comments. Your revised manuscript is also likely to be sent to reviewers for further evaluation.

Sincerely,

Jose Luis Balcazar, Ph.D.

Academic Editor

PLOS Pathogens

D. Scott Samuels

Section Editor

PLOS Pathogens

Michael Malim

Editor-in-Chief

PLOS Pathogens

orcid.org/0000-0002-7699-2064

Reviewer's Responses to Questions

**Part I - Summary**

Reviewer #1: Very nice work. I suggest "accept" upon the satisfaction of several minor suggestions.

Reviewer #2: In this study, Nashier et al. show that functional conservation of the HipA/B operon in Klebsiella pneumoniae. By overexpressing HipA, they denmonstrate that HipA overexpression is toxic, and the toxicity of HipA overexpression is rescued by the overexpression of HipA antitoxin HipB. Notably, the overexpression of HipA led to tolerance to the antibiotic ciprofloxacin, which is commonly used for the treatment of Klebsiella pneumoniae. To illuminate the role of HipA in antibiotic tolerance, the authors used quantitative mass spectrometry to comprehensively identify changes in proteome and phosphoproteome. These analyses confirmed the kinase activity of HipA, and importantly, identification of many dozens of HipA substrates. These high-confidence HipA substrates are involved in diverse biological pathways, implying a role of HipA in their regulation.

Understanding the mechanisms of antibiotic resistance is of major importance, and this work provides a major resource for understanding the role of HipA-catalyzed phosphorylation in antibiotic resistance. Technically, mass spectrometry analyses are performed at impressive depth and the results are highly reproducible. I believe that work will be of broad interest to researchers across fields.

Reviewer #3: The manuscript „Deep phosphoproteomics of Klebsiella pneumoniae reveals HipA-mediated tolerance to ciprofloxacin” by Nashier et al. presents the results of the proteomic and phosphoproteomic study of the HipA protein in Klebsiella pneumoniae, which by homology to the E.coli was predicted to be a kinase. The authors show that HipA overexpression indeed leads to phosphorylation changes, and the datasets comparing bacteria overexpressing HipA to wild type and the HipA overexpressors exposed to antibiotics to those that weren’t provide a valuable resource for further studies. The presented data are solid but there are a few issues that should be resolved before publication.

Reviewer #4: The study investigates phosphorylation of HipA-associated proteins in the globally important bacterial pathogen, Klebsiella pneumoniae. The study is well presented and well written; however, the text switches between experiments with Klebsiella and E. coli, which is challenging for the reader. The study is highly descriptive for the phosphorylation studies, it would be nice to see one step further by disruption of a phospho site and the impact. Additionally, key information on the number of replicates for total proteome and phosphoproteome analysis is needed, as well as clarification and robustness in data analysis for normalization, false discovery rate correction, and phosphosite localization probability do not appear to be included. These analyses could drastically change the findings of the paper.

**Part II – Major Issues: Key Experiments Required for Acceptance**

Reviewer #1: NA

Reviewer #2: I have a few comments that should be addressed before the manuscript is accepted for publication.

1. To directly demonstrate a role of HipA kinase activity in Klebsiella pneumoniae , it would be useful to show that kinase dead mutant of HipA is not toxic to Klebsiella pneumoniae, and unable to provide resistance to ciprofloxacin.

2. Perhaps I missed it, but it is unclear to me whether there is any data about the expression of HipA/B in Klebsiella pneumoniae isolated from antibiotic-sensitive and anti-biotic resistanace patients?

Reviewer #3: Detailed comments:

1. General: the increase in phosphorylation of the proteome when HipA is overexpressed is not a direct proof that it is, in fact, a kinase. There is a lot of indirect evidence pointing to that but this shortcoming should be discussed by the authors.

2. The authors note the slower growth and lower viability in K. pneumoniae overexpressing HipA, and attribute it to the toxicity of HipA. The same (slow growth) was observed in E.coli, is that also because HipA from K. pneumoniae is toxic to E. coli, or overexpression of E.coli HipA in E.coli would have the same effect?

3. Overexpression of HipA results in HipA phosphorylation on S150 and T158 both in E.coli and K. pneumoniae. Are the phosphopeptides unique to K. pneumoniae HipA? Does it mean that HipA is constitutively autophosphorylated? Can these phosphosites be detected in WT (which, after all, expresses HipA) using targeted proteomics?

4. If HipA overexpression leads to ciprofloxacin resistance, is the reverse also true (is HipA overexpressed in the resistant strains)?

5. The authors speculate (line 344 and the subsequent paragraph) that numerous pathways are potentially post-translationally regulated by HipA. This statement can be easily backed up by normalizing the phosphoproteomics to proteomics – are the increases in phosphorylation of specific sites independent of the increase in protein levels? (moreover, both proteome and phosphoproteome changes could be compared to the transcriptome profiles, but this would of course require additional experiments so may be beyond the scope of the current work).

Reviewer #4: Specific comments

- In the Results, the flow of information is logical and well-presented but the switch from Klebsiella in 1.1 to E. coli 1.2 and then back to Klebsiella in 1.4 is not intuitive. If the E. coli overexpression of hipA is known, should section 1.2 and 1.3 be presented in supplemental or include Klebsiella within these sections? If hipA is well characterized in E. coli and the point of the manuscript is to provide novel information for Klebsiella, this point is diluted with the current approach.

- Was the phosphoproteome data normalized to the total proteome? If not, then a consideration to possible elevated overproduction in the total proteome influencing the phosphoproteome levels should be presented. Is this what line 632 refers to?

- Line 549: how many biological replicates were used?

- There are limitations of identifying phosphosites in Klebsiella using TiO2 beads References: https://pubmed.ncbi.nlm.nih.gov/25156620/; https://pubmed.ncbi.nlm.nih.gov/30231186/; https://pubmed.ncbi.nlm.nih.gov/29377012/). Were the authors able to detect phosphohistidine and phosphoaspartate residues with the acidified samples? Re-searching the total proteome data bioinformatically within these as modifications may provide a new level of insight into the study.

- For robustness in data analysis, significantly different sites should be determined following a statistical test (e.g., Student’s t-test, Welch’s t-test) and multiple hypothesis testing correction with a false discovery rate at 5% or less. From line 644, was this correction performed? If not, it should be and data should be presented as those that reach the FDR and those that do not.

- For robustness in data analysis, localization probability of >75% is needed for accurate phosphopeptides identifications. Was this performed? It is not clear from the methodology.

- Fig. 5f – great to see the comparison to other studies; however, this figure is misleading as the same statistical testing and robustness was not presented in the current study. This needs to be completed for an accurate comparison.

**Part III – Minor Issues: Editorial and Data Presentation Modifications**

Reviewer #1: (1) I do not find the figure legends.

(2) Line 636, "Re-quantify" should be modified to the noun form.

Reviewer #2: None

Reviewer #3: (No Response)

Reviewer #4: - Add italics and proper formatting to scientific names within the references.

- Fig. 2A – plots are not clear as to how they represent growth curves. Suggest revising. Connection to 2B is unclear. Same for Fig. 3A. What value do these plots add? Fig. 3B is a clear growth curve.

- Fig. 2C and D, 3C and D, and 4C and D – why not present as volcano plots with complete statistical power?

- Growth curves for Fig. 4?

- The figure legends are within the main text but without the figure itself – suggest revising.

- Supp. Fig. 2 – are these representative of volcano plots? Why present in this manner?

- The PRIDE data set is not publicly available and therefore, the login information and password for reviewers is needed.

PLOS authors have the option to publish the peer review history of their article (what does this mean?). If published, this will include your full peer review and any attached files.

Reviewer #1: No

Reviewer #2: No

Reviewer #3: No

Reviewer #4: **Yes: **Jennifer Geddes-McAlister
---

## [Decision Letter · Decision Letter 1]

19 Nov 2024

Dear Prof. Macek,

We are pleased to inform you that your manuscript 'Deep phosphoproteomics of Klebsiella pneumoniae reveals HipA-mediated tolerance to ciprofloxacin' has been provisionally accepted for publication in PLOS Pathogens.

Best regards,

Jose Luis Balcazar, Ph.D.

Academic Editor

PLOS Pathogens

D. Scott Samuels

Section Editor

PLOS Pathogens

Michael Malim

Editor-in-Chief

PLOS Pathogens

orcid.org/0000-0002-7699-2064

The authors have addressed all reviewers' comments and made substantial improvements to the manuscript. Many thanks!

Reviewer Comments (if any, and for reference):

Reviewer's Responses to Questions

**Part I - Summary**

Reviewer #1: The revised version satisfied all my concerns and it can be accepted.

Reviewer #2: The manuscript provides an importance, high-quality resource for understanding bacterial resistance. I believe that this resource will catalyze further research and contribute to understanding the mechanisms of bacterial drug tolerance.

Reviewer #3: The authors addressed all my concerns in a very meticulous and detailed way. Im my opinion, the study is now ready for acceptance.

Reviewer #4: (No Response)

**Part II – Major Issues: Key Experiments Required for Acceptance**

Reviewer #1: na

Reviewer #2: None.

Reviewer #3: none

Reviewer #4: The authors have done a very thorough job of addressing each of my comments and suggestions. I am impressed with the attention to detail and the comprehensive re-analyses performed to address the comments and strengthen the manuscript. It is worth noting that the number of biological replicates only be two or three is quite low for a phosphoproteomics experiment with the field standard of five biological replicates for phosphoproteomics experiments; however, the authors have used appropriate controls and statistical robustness within the revised manuscript that provide the best analysis given the limited number of replicates.

**Part III – Minor Issues: Editorial and Data Presentation Modifications**

Reviewer #1: na

Reviewer #2: The revised manuscript is greatly improved, and I fully endorse its publication in PLOS Pathogens.

Reviewer #3: none

Reviewer #4: (No Response)

PLOS authors have the option to publish the peer review history of their article (what does this mean?). If published, this will include your full peer review and any attached files.

Reviewer #1: No

Reviewer #2: No

Reviewer #3: No

Reviewer #4: **Yes: **Jennifer Geddes-McAlister

---

## [Editor Report · Acceptance letter]

25 Nov 2024

Dear Prof. Macek,

We are delighted to inform you that your manuscript, "Deep phosphoproteomics of Klebsiella pneumoniae reveals HipA-mediated tolerance to ciprofloxacin," has been formally accepted for publication in PLOS Pathogens.

Best regards,

Michael Malim

Editor-in-Chief

PLOS Pathogens

orcid.org/0000-0002-7699-2064